# Analyses of the Genetic Diversity and Population Structures of *Sporothrix* spp. Clinical Isolates from Paraíba, Brazil

**DOI:** 10.3390/jof10120848

**Published:** 2024-12-09

**Authors:** Larissa Alves da Silva, Jamile Ambrósio de Carvalho, Luanna de Oliveira e Lima, Francisco Bernardino da Silva Neto, Edeltrudes de Oliveira Lima, Walicyranison Plínio da Silva Rocha, Zoilo Pires de Camargo, Anderson Messias Rodrigues, Ana Carolina Bernardes Dulgheroff, Felipe Queiroga Sarmento Guerra

**Affiliations:** 1Bioactive Natural and Synthetic Products Postgraduate Program, Laboratory of Mycology, Federal University of Paraiba (UFPB), João Pessoa 58051-900, Brazil; luanna@ltf.ufpb.br (L.d.O.e.L.); edelolima@yahoo.com.br (E.d.O.L.); 2Laboratory of Emerging Fungal Pathogens, Department of Microbiology, Immunology, and Parasitology, Discipline of Cellular Biology, Federal University of São Paulo (UNIFESP), São Paulo 04023-062, Brazil; jamileambrosio@hotmail.com (J.A.d.C.); zpcamargo@unifesp.br (Z.P.d.C.); amrodrigues.amr@gmail.com (A.M.R.); 3Lauro Wanderley University Hospital, Federal University of Paraiba (UFPB), João Pessoa 58051-900, Brazil; fbsnneto@gmail.com; 4Department of Infectious, Parasitic and Inflammatory Diseases, Center for Medical Sciences, Federal University of Paraíba (UFPB), João Pessoa 58051-900, Brazil; 5Laboratory of Mycology, Department of Pharmaceutical Sciences, Federal University of Paraiba (UFPB), João Pessoa 58051-900, Brazil; wps@academico.ufpb.br (W.P.d.S.R.); fqsg@academico.ufpb.br (F.Q.S.G.); 6National Institute of Science and Technology in Human Pathogenic Fungi, São Paulo 04023-062, Brazil; 7Professional and Technological Centre, Technical School of Health, Federal University of Paraiba (UFPB), João Pessoa 58051-900, Brazil; anadulgheroff@gmail.com

**Keywords:** sporotrichosis, identification, molecular, *Sporothrix brasiliensis*, epidemiology, morphology, species-specific

## Abstract

Sporotrichosis is a subcutaneous mycosis of global distribution, capable of affecting both humans and animals, and caused by species of the genus *Sporothrix* spp. This study aimed to evaluate the genetic diversity and mating type distribution of clinical isolates of human sporotrichosis in Paraíba, Brazil, to better understand the population structure, epidemiology, and diversification of this pathogen, as well as to explore possible transmission routes. Methods: A total of 36 clinical isolates were morphologically identified, and clinical and demographic data were collected. Fungal DNA extraction was then performed, followed by species-specific PCR using markers targeting the calmodulin gene. The mating type idiomorph of the species was identified by PCR using primers targeting the MAT1-1 and MAT1-2 loci. Amplified Fragment Length Polymorphism (AFLP) was used to evaluate the genetic variability of *Sporothrix* spp. Results: The distribution of the disease identified that all cases occurred in João Pessoa and adjacent cities. From the 36 isolates, the majority (75%) being affected females, a prevalent occurrence of the lymphocutaneous form, and 98% zoonotic transmission were confirmed. Micro- and macromorphological structures were similar to each other, confirming *Sporothrix* spp. All isolates were confirmed as *S. brasiliensis* and the presence of a single sexual idiomorph, MAT1-2, was detected. The AFLP results indicate the possibility of the circulation of one or two genetic groups in João Pessoa and the metropolitan region. Conclusions: To our knowledge, this is the first time isolates in the Paraíba state are genetically characterised, all identified as *Sporothrix brasiliensis*. It is likely that this species in Paraiba originated from Rio de Janeiro, as all they possess the MAT1-2 idiomorph, indicating low intergenotypic variation.

## 1. Introduction

Sporotrichosis is a fungal disease, which can be either acute or chronic. It is caused by thermodimorphic fungi and belongs to the clinical clade of *Sporothrix* spp. It affects humans and animals (primarily) through traumatic implantation of the fungus into the skin. The “pathogenic clade” includes the following species: *Sporothrix schenckii* (*sensu stricto*), *S. brasiliensis*, *S. globose*, and *S. luriei* [1,2].

The classic transmission of sporotrichosis is sapronotic, via the traumatic implantation of fungal propagules into subcutaneous tissue, is often related to contact with contaminated plants and soil. It is also commonly associated with occupational or recreational activities such as gardening, floriculture, and agriculture. However, zoonotic transmission has recently advanced, particularly with the involvement of cats in the sporotrichosis transmission route [3,4].

Sporotrichosis is a neglected disease and is not a notifiable disease in all regions where it occurs, so its exact prevalence is unknown. Since 1998, Brazil has experienced an advance of the disease, particularly in the southeast, with the state of Rio de Janeiro and Rio Grande do Sul reporting the highest number of cases [5,6]. Recently, there has also been an increase in the incidence of sporotrichosis outbreaks in cats and humans in the northeastern region of Brazil, particularly in Rio Grande do Norte and Pernambuco, states neighbouring Paraíba [7,8].

Laboratory diagnosis of infection is usually performed by isolating and identifying the microorganism using culture (gold standard), since direct mycological or histopathological examination of human clinical samples presents low sensitivity. However, the identification of species in the clinical clade of *Sporothrix* by phenotypic methods is unreliable, necessitating an alternative, such as molecular analysis by species-specific Polymerase Chain Reaction (PCR) [6,9].

In Paraiba, human sporotrichosis became a notifiable disease in 2018. According to the epidemiological bulletin from the State Health Department of Paraiba, the number of notifications per year has been increasing, with 711 cases reported in 2019 and 577 cases in 2023 until week 44. Therefore, it has become essential to investigate the epidemiological characteristics of the mycosis in these cases and the species distribution involved, since there are currently no molecular data on *Sporothrix* spp. in the state of Paraiba.

Additionally, understanding the phylogenetic variability of *Sporothrix* spp. is crucial, as it may provide insights into potential mutation rates in the fungal progeny. Given the importance of the *Sporothrix* genus as an infection causing agent in humans, and the lack of clinical-epidemiological data in Paraiba (Brazil), this study aimed to capture the distribution and biological behavior of the *Sporothrix* species involved in these cases.

## 2. Materials and Methods

### 2.1. Ethics Statement

The authors confirm that the ethical policies of the journal, as noted on the journal’s author guidelines page, have been adhered to and the appropriate ethical review committee approval has been received. All clinical and demographic data of the patients were collected in accordance with the Local Research Ethics Committee from the Federal University of Paraiba, approved under number 65629622.0.0000.5188. The written consent was waived because of data anonymization and commitment to preserve the identity of the patients.

### 2.2. Sporothrix Isolates

A total of 36 *Sporothrix* isolates from clinical origin were included in this study. The clinical isolates were retrieved from patients with clinically suspected sporotrichosis attending the University Hospital Lauro Wanderley (HULW), Federal University of Paraiba (UFPB), between 2019 and 2020. The isolates were grown on Sabouraud dextrose agar slants (SDA; Difco Laboratories, Detroit, MI, USA) for 7–14 days at 25 °C. Colonies that exhibited morphological aspects similar to *Sporothrix* sp. were subcultured and stocked in glass tubes with water supplemented with chloramphenicol at room temperature for further identification [10]. The clinical data from the patients were collected from their respective medical records within the SM—ULAC (Mycology Sector—Clinical Analysis Laboratory Unit) at HULW, following the hospital’s ethical standards.

### 2.3. Morphological Characterisation of Sporothrix Isolates

*Sporothrix* isolates were identified according to the established methods described by Marimon et al. (2007) [11]. Characteristics of vegetative and reproductive mycelium were analysed in micromorphology (presence of hyphae, pigmentation, and arrangement of conidia) and macromorphology (diameter, texture, and colour of colonies). Morphological characters were evaluated from cultures grown on Potato Dextrose Agar (PDA) (200 g potato, 15 g dextrose, 15 g agar, and 1 L distilled water) at 25 °C for 14 days. For the microscopic characterisation of the isolates, the following were studied: the presence or absence of sessile conidia, the shape, colouring, ornamentation, arrangement of the conidia using the Ridell (1950) microculture technique [12]. After growth, it was read using a Nikon Eclipse Ci^®^ optical microscope (Melville, NY, USA).

### 2.4. Molecular Identification

#### 2.4.1. Extraction of Genomic DNA

Genomic DNA was extracted and purified directly from 14-day-old monosporic colonies grown on Sabouraud Dextrose Agar (SDA) (Difco™ BD/Sparks, MD, USA) using the Fast DNA kit (MP Biomedicals, Vista, CA, USA) [13]. DNA concentrations were estimated on a NanoDrop 2000 spectrophotometer (ThermoFisher Scientific, Wilmington, DE, USA) and stored at −20 °C until use) [14].

#### 2.4.2. Species-Specific PCR

All DNA extracted from the isolates was characterised to the species level, using species-specific Polymerase Chain Reaction (PCR) assay that targets fragments of the calmodulin gene) [14]. Reference strains of (*S. brasiliensis*—Ss08) and the negative control (absence of DNA) were used as control samples. PCR products were analysed by 1.2% agarose gel electrophoresis at 100 V for 1 h in the presence of GelRed (Biotium, Hayward, CA, USA). Bands were visualised with UV light equipment L-Pix (Loccus Biotecnologia, São Paulo, Brazil) [14].

#### 2.4.3. PCR Detection of Mating-Type (MAT) Gene

The extracted DNA was also used to perform selective PCR for the MAT locus using primers for the MAT1-1 and MAT1-2 targets described by Carvalho et al. (2021) [15]. Reference strains of each corresponding idiomorph were used as positive controls. Amplicons of 673 pb were observed for isolates MAT1-1 and 291 pb for MAT1-2.

#### 2.4.4. Amplified Fragment Length Polymorphism (AFLP) Fingerprinting

The extracted DNA was purified using a genomic DNA purification kit to perform AFLP using the protocol by Vos et al. (1995), with modifications [16]. Briefly, *Sporothrix* genomic DNA (200 ng) was digested using the restriction enzymes EcoRI (GAATTC) and MseI (TTAA) and connected to the EcoRI and MseI adapters simultaneously. A pre-selective PCR was performed with the primers EcoRI + 0 and MseI + 0 [16]. Finally, a selective PCR was performed using the EcoRI Primer with two base selection (5′-GAC TGC GTA CCA ATT CNN-3′) labelled with 6-carboxyfluorescein (FAM; blue) and the unlabelled MseI primer with selection of two bases (5′-GAT GAG TCC TGA GTA ANN-3′). Two combinations were used for genotyping *Sporothrix* isolates (#3 EcoRIGA/MseI-TT and #5 EcoRI-GA/MseI-AG). AFLP fragments were resolved by capillary electrophoresis with an ABI3730xl genetic analyser alongside a GeneScan LIZ600 Internal Size Standard (35–600 bp; Applied Biosystems Foster City, CA, USA). To evaluate the ability to accurately reproduce results, electropherograms were representative of two independent assays. Fragment analysis was performed using the BioNumerics software. v.7.6 (Applied Maths, Sint-Martens-Latem, Belgium).

#### 2.4.5. Analysis of AFLP Data

In relation to the raw data generated by AFLP, only strong and high-quality fragments with sizes between 50 and 500 base pairs were considered. Peak patterns were converted into a binary matrix considering the presence (1) or absence (0) of fragments in certain positions, as described by Carvalho et al. (2020) [5]. From these data, genetic distance was calculated using Jaccard’s similarity coefficient and dendrograms were constructed using the unweighted pairwise average arithmetic method (UPGMA) in order to verify the relationship between the different *Sporothrix* species. Also to evaluate the diversity and evolutionary relationship between species, the minimum spanning tree model (MST), principal component analysis (PCA), multidimensional scaling (MDS) and self-organizing map (SOM) were performed as described by Carvalho et al. (2020) and Carvalho et al. (2021b) [5,15].

Basically, the MST provides an indication of evolutionary directionality, which can even be used to understand the transmission of pathogens. It is a tree that connects all samples to minimize branch distances, the oldest isolates are in the middle and the younger at terminal nodes [17]. Reduction and dimensional methods (PCA and MDS) were used to generate three-dimensional graphs that represent how isolates are dispersed according to their similarity. And finally, the SOM is an artificial learning neural network that also organizes based on similarity, but in a two-dimensional way (map), the lighter shades of grey and thicker lines indicate greater genetic distance between the isolates [6,18]. All figures were generated in Corel Draw X8 (Corel, Ottawa, ON, Canada).

## 3. Results

### 3.1. Population Distribution

According to our results, *Sporothrix brasiliensis* was the only species identified, both in metropolitan regions and city neighbourhoods. It should be noted that these data only include patients treated at the Lauro Wanderley University Hospital, by the infectious disease service, which is a reference center for sporotrichosis.

The distribution of the disease, according to the residence of the infected individuals, revealed that all of the cases occurred in urban areas. Twenty-two patients resided in the city of João Pessoa, state of Paraíba. Of these cases, 12 occurred in the southern part of the city, with the highest concentration in the Valentina neighbourhood. This was followed by five cases in the western part of the city, while the eastern and northern city sectors presented three and two cases, respectively. Additionally, 14 cases occurred in neighbouring municipalities, with Bayeux presenting the highest rate, followed by Santa Rita, Conde, and Jacumã. These data are illustrated in Figure 1.

### 3.2. Epidemiological and Clinical Characteristics of the Population

Based on the data collected from the patients included in this investigation, shown in Table 1, the majority were female (*n* = 27, 75%) and 25% (*n* = 9) were male. Regarding clinical forms, all cases presented a history of cutaneous sporotrichosis, located on the lower or upper limbs, in either the fixed cutaneous or lymphocutaneous form, with the latter form being more prevalent, representing 69% of the cases. Detailing the source of infection, it was found that most cases involved reports of accidents with animals, principally infected cats, characterising a zoonotic transmission route.

A time variation between the onset of the first symptoms and the definitive diagnosis, ranging from 1 week to 4 months was noted. As illustrated in Figure 2, most patients (*n* = 23) were diagnosed between 3 weeks and 1 month after the appearance of initial signs and symptoms.

### 3.3. Macromorphology and Micromorphology

Based on macro and micromorphological characteristics, the 36 isolates were identified as *Sporothrix* spp. The strains replicated on PDA medium at 25 °C for 21 days. Macromorphological characteristics of the colonies varied in pigmentation and melanin production. These were divided into two categories: black and mixed. Black colonies predominated, comprising 58% (21) of the samples, displaying black frontal sides with intense melanin production. However, some strains presented mixed characteristics, exhibiting a combination of black and white colours. The colonies reached a diameter of 33 to 51 mm; all presented slightly wrinkled surfaces, defined zones, and beige to transparent peripheries (Figure 3).

Micromorphological analyses revealed similar arrangements. Hyaline, septate, and branched hyphae were observed, along with clusters of small, hyaline sympodial conidia at the ends of the conidiophores, and pigmented sessile conidia, which were brown to dark brown in colour, with thick walls, being globose to subglobose in shape. The average area of the conidia was from 5.78 to 3.38 μm, with the pigmented conidia presenting a larger area compared to the hyaline conidia present in the analysed slides (Figure 3).

### 3.4. Species and Mating-Type Idiomorph Identification

For species-level identification, the isolates 36 were evaluated using species-specific PCR. All samples were identified as *S. brasiliensis*. Additionally, mating type was verified through selective PCR for mating type genes, represented by the idiomorphs MAT1-1 or MAT1-2. The results indicated the presence of a single idiomorph (MAT1-2) in each tested isolate.

Typical AFLP dendrograms based on Jaccard’s similarity coefficient were used to assess the diversity within the *Sporothrix brasiliensis* isolates samples. Two markers were employed to verify the genetic variation in the agents (#3 EcoRI-GA/MseI-TT and #5 EcoRI-GA/MseI-AG). In total, 178 fragment points were amplified in the range of 50 to 500 bp, as shown in Figure 4 and Figure 5. After analysing the AFLP results, three main groups were observed in both dendrograms for the respective markers #3 EcoRI-GA/MseI-TT and #5 EcoRI-GA/MseI-AG: group I (JS = 25.354% ± 2.38%) and (JS = 24.787% ± 2.25%) consisting of reference and clinical strains of *S. brasiliensis*; group II (JS = 18.980% ± 1.57%) and (JS = 25.393% ± 2.33%) containing reference isolates of *S. schenckii*, and group III (JS = 27.900% ± 2.57%) and (JS = 23.247% ± 5.14%) formed by reference strains of *S. globosa*. High cophenetic correlation coefficients (≥99%) were observed in the branches separating these groups.

In the #3 EcoRI-GA/MseI-TT dendrogram (Figure 4), group I isolates are divided into three subclades (Ia, Ib, and Ic). Subclade Ia (JS = 71.640% ± 4.99%) consists exclusively of clinical isolates from Paraiba, indicating the circulation of a single genetic group in the state. Subclades Ib (JS = 66.680% ± 1.77%) and Ic (JS = 70.500% ± 0.00%) comprise the reference strains from various regions of Brazil. In the #5 EcoRI-GA/MseI-AG combination (Figure 5), group I was divided into four subclades (Ia, Ib, Ic, and Id). Subclades Ia (JS = 78.487% ± 5.75%) and Ib (JS = 79.980% ± 0.00%) consist exclusively of clinical isolates from João Pessoa and adjacent cities, indicating the circulation of at least two genetic groups in this region. Subclades Ic (JS = 75.047% ± 0.42%) and Id (JS = 69.747% ± 2.60%) are formed by reference strains from various regions of Brazil.

The marker characteristics for the primer combinations used in the AFLP reaction are provided in Appendix A, which revealed the excellent ability of each primer combination to detect intra- and interspecific polymorphisms.

Cluster analysis performed based on MST (Figure 6) identified evidence of genotypic dispersion in João Pessoa and metropolitan regions, where the main genetic groups circulating in recent outbreaks transmitted by cats are related to genotypes of *S. brasiliensis* that occur in the states of Rio de Janeiro and Pernambuco. This proximity can also be seen in the dendrograms #5 EcoRI-GA/MseI-AG. Next, the population structure was evaluated using multivariate analyses such as Principal Component Analysis (PCA) and Multidimensional Scaling (MDS), represented by graphs in Figure 7. It was possible to visualise the subdivision of genetic groups formed in both graphs, and a small intraspecific genetic diversity among the isolates from Paraíba. The PCA combination (#3 and #5) revealed the cumulative percentage sum of the first three components (X, Y, and Z), which was 59.1% for both, indicating a robust genetic structure.

Analysis of the SOM results (Figure 8A) shows neural networks where the presence of reconstructed scenarios in *S. brasiliensis* genotypes can be observed, and also indicates little intraspecific diversity between isolates, which is characteristic of outbreaks from a common source, such as occurs in the transmission of sporotrichosis by cats. On the other hand, genetic distance between species can also be observed.

## 4. Discussion

Regarding the epidemiological profile of the patients included in this study, the majority were female, similar to the findings of an outbreak in southern Brazil [19]. Despite this mycosis affecting people regardless of age and sex, occupational activities and animal contact influence the infection rate in distinct populations. Studies indicate a high incidence among women due to more contact with felines, which are currently the primary transmission source of infection [20,21].

The main clinical presentations and their anatomical locations in the isolates of this study are consistent with the literature, revealing higher incidences on the lower and upper limbs, which are more exposed areas, and at risk of fungal inoculation through trauma, resulting primarily in cutaneous lesions, with or without lymphatic involvement [19]. From the data collected, a prolonged period between lesion onset and diagnosis of up to 4 months was observed. This suggests that sporotrichosis continues as a neglected disease, and despite diagnostic advances, immediate clinical suspicion is often lacking due to limited specific information concerning these infections [4,22].

Cases of human sporotrichosis have been reported in nearly all 26 Brazilian states and are directly related to feline *S. brasiliensis* transmission, predominantly concentrated in the southwest and southern regions of the country [23]. However, the disease is increasingly spreading in the Brazilian northeast [5].

The capital of Paraíba, João Pessoa, is divided into four zones: south, east, north, and west. Of these zones, the southern zone stands out for its high number of reported cases. This is followed by the metropolitan regions, which may be partly explained by urban expansion. Characteristically, this is a diverse environment comprising areas engaged in both rural and urban activities in a single space. Alzuguir et al. (2020) have also noted a higher incidence of the disease in Rio de Janeiro, in areas with low per capita income and inadequate sanitation infrastructure [24].

In this study, socioeconomic characteristics were not studied by locality, but a higher number of cases were observed in regions presenting recent urbanization. During mating season, cats can roam up to 6 km, transmitting disease to other neighbourhoods. This behavior may be responsible for the spread and explain the occurrence of cases in adjacent areas [10,25,26].

The phenotypic characterisation of *Sporothrix* spp. isolates in this study revealed an average colony growth of 39 mm, with dark brown to black pigmentation associated with melanin production in the fungal cell structure. One hypothesis for this dark colouration could be the warmer climate in this region of Brazil, where increased temperatures may have prompted the fungus to develop defense mechanisms to enhance its thermotolerance. Melanin is an important virulence factor, crucial for the organism’s survival within the environment and against aggressions it might encounter as well [27].

Microscopic analysis revealed abundant production of pigmented sessile conidia in all 36 strains, displaying similar morphology among them. Septate and branched hyphae were observed, with conidiophores producing pear-shaped/oval hyaline conidia, along with globose or subglobose conidia with dark pigmentation indicating the presence of melanin. These characteristics are consistent with previous studies associating *S. brasiliensis* with higher quantities of dematiaceous globose conidia [11,28].

In human sporotrichosis diagnoses, mycological culture is considered the gold standard, despite specific limitations. *Sporothrix* species are morphologically and physiologically very similar, making phenotypic identification imprecise, with identification periods (depending on fungal colony growth) ranging from 5 to 10 days. These factors directly impact the need for early diagnosis and epidemiological control of species circulating within Brazil [29,30]. Molecular diagnosis plays a crucial role in managing treatment, reducing identification time, and providing more accurate results. It is also epidemiologically important for recognizing outbreaks involving the varied *Sporothrix* species [6]. All samples in this study were identified through species-specific PCR as *S. brasiliensis*, which as been the most commonly isolated species in Brazil for feline sporotrichosis since 2013. *S. brasiliensis* is the most virulent species in the complex, associated with more severe clinical presentations in humans and infections in animals, from which zoonotic transmission occurs. Moreover, in Brazil, *S. brasiliensis* is considered the center of dispersion, and generally presents better responses to antifungals (both in vitro and in vivo) when compared to other species [6,31,32,33,34].

The results of this study provide insights into the genetic diversity and population structure of *Sporothrix brasiliensis* in greater João Pessoa and its metropolitan region. The finding of only *S. brasiliensis*, MAT1-2, and the analysis of AFLP fragments revealing low intra-specific diversity suggest a single transmission route in the region, likely being predominantly clonal, unlike isolates from Espírito Santo, where samples of *S. schenckii* and both idiomorphs were found [35].

Furthermore, the origin of the samples found in Paraíba is likely Rio de Janeiro, since when analysing the infection routes of *S. brasiliensis* in Brazil, two main types are observed: MAT1-1, prevalent in the states of Minas Gerais and Rio Grande do Sul, and the rest, which belong to Rio de Janeiro, where idiomorph MAT1-2 prevails [36]. This suggests that due to the high frequency of MAT1-2 loci, the *S. brasiliensis* population migrated from Rio de Janeiro (the epicenter of sporotrichosis outbreaks in Brazil) to João Pessoa and neighbouring cities [36]. Therefore, the low intra-specific diversity found among the samples in this study and the probable origin and migration direction from southeast to northeast are suggestive of a founder effect described by Carvalho et al. (2021) in other samples from the northeast region. This finding aids in understanding the dynamics of *S. brasiliensis* expansion in Brazil [5].

The effect is explained by the hypothesis that a parental genotype, in this case from Rio de Janeiro, migrates to a particular region, leading to direct horizontal transmission between animals, and establishing and generating a founder population. The result is fewer circulating genotype variations in the region, as observed in the isolates of this study [15,37]. Thus, there is a need to establish sanitary barriers to contain greater dissemination. Further studies, including samples from other cities in the state of Paraíba farther from the capital, may find *S. brasiliensis* isolates with greater intra-specific diversity.

Despite the close genetic relationship of the isolates found in Paraíba with samples from Rio de Janeiro and the neighbouring Pernambuco, they do not cluster in the same clade. According to Thines (2019), diversification through host shifts is generally followed by radiation, specialization, and speciation, a result of the evolutionary process of effective adaptation to the new host [38].

In addition, the characteristics of the markers highlighted in Appendix A confirm the low diversity of *S. brasiliensis*, as indicated by the statistical values provided by AFLP. The data show that the primers performed well, as observed by Carvalho et al. (2021) [15].

A high genetic diversity was found among species, as evidenced more clearly in SOM. *S. schenckii* exhibits greater genetic separation, consistent with other studies which report *S. schenckii* as having the highest genetic diversity among *Sporothrix* species [5,18,37].

Considering that the study samples were of convenience, covering only the capital and metropolitan regions and the years of 2019 and 2020, future studies that expand the coverage area and period of the study would be important to consolidate this genetic characterisation of *Sporothrix* in Paraíba.

## 5. Conclusions

In this study, clinical data revealed that sporotrichosis follows a pattern of zoonotic transmission, primarily through cats. Due to occupational activities, women are the most affected, showing higher rates of lymphocutaneous and fixed cutaneous involvement. Additionally, demographic data suggest an urban epidemic profile in the Paraíba region.

Micro- and macromorphological criteria enabled identification of the disease-causing genus; however, in species identification, molecular tools proved to be more precise.

This study characterises genetically, for the first time, the *Sporothrix* isolates from Paraíba. The only idiomorph found was MAT1-2, with low intra-genotypic variation observed compared to *S. brasiliensis* samples from other states, suggesting a founder effect in the Paraíba isolates, likely originating from Rio de Janeiro. We therefore see the need to implement sanitary barriers, as well as molecular diagnostic tools to effectively control and mitigate the disease.

## Figures and Tables

**Figure 1 jof-10-00848-f001:**
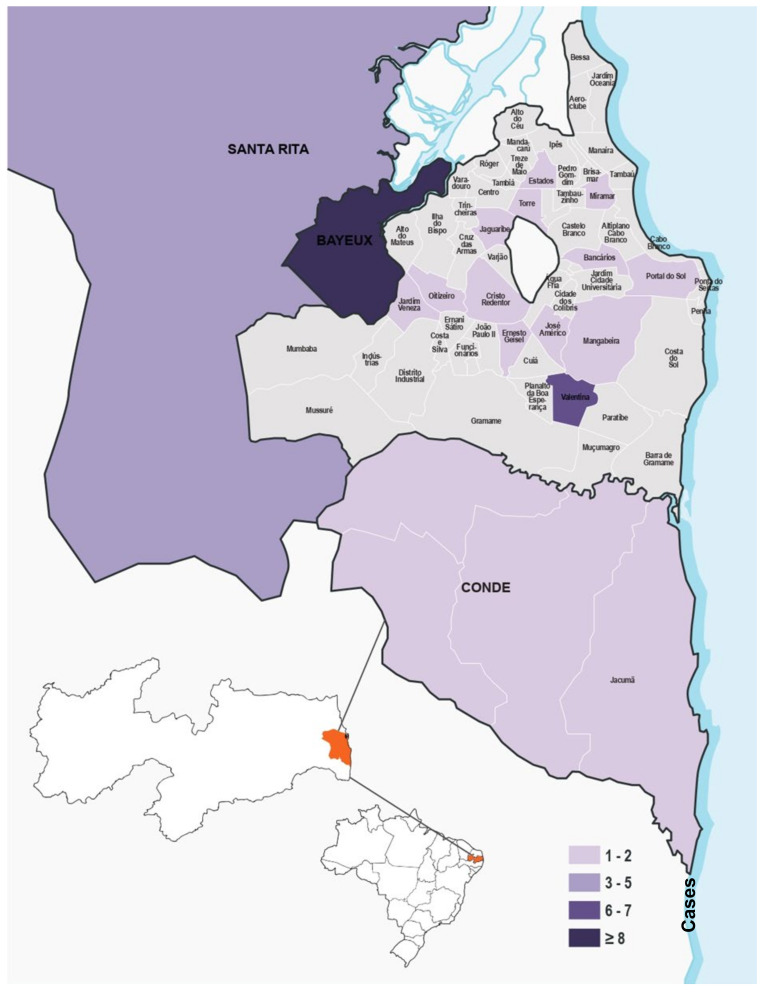
Geographic distribution and prevalence of *Sporothrix* species, between 2019 and 2020 in the city of João Pessoa-PB (*n* = 36).

**Figure 2 jof-10-00848-f002:**
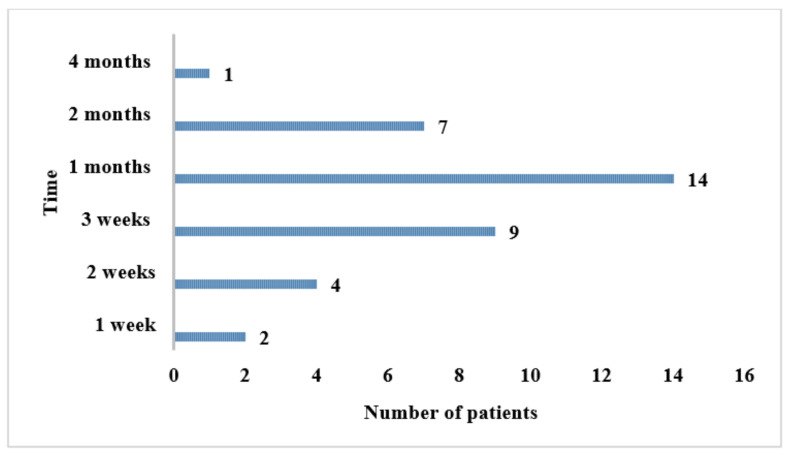
Time between the onset of the first symptoms of sporotrichosis and the definitive diagnosis in study patients (*n* = 36).

**Figure 3 jof-10-00848-f003:**
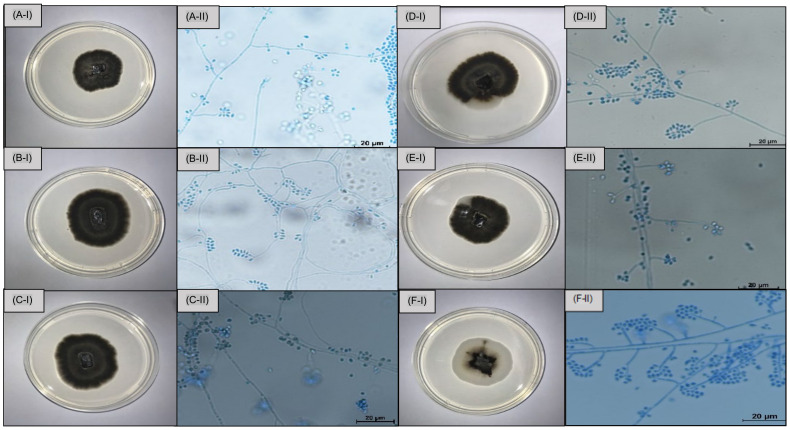
Macro and micromorphology of the *Sporothrix brasiliensis* identified in the study cultivated on PDA at 25 °C for 14 days. (**A-I**–**C-I**): black pigmented colonies. (**A-II**–**C-II**): hyaline hyphae containing some dematiaceous conidia. (**D-I**–**F-I**): holonies with mixed pigmentation. (**D-II**–**F-II**): hyaline hyphae containing some dematiaceous conidia. Scale bars: 20 µm. No colouring.

**Figure 4 jof-10-00848-f004:**
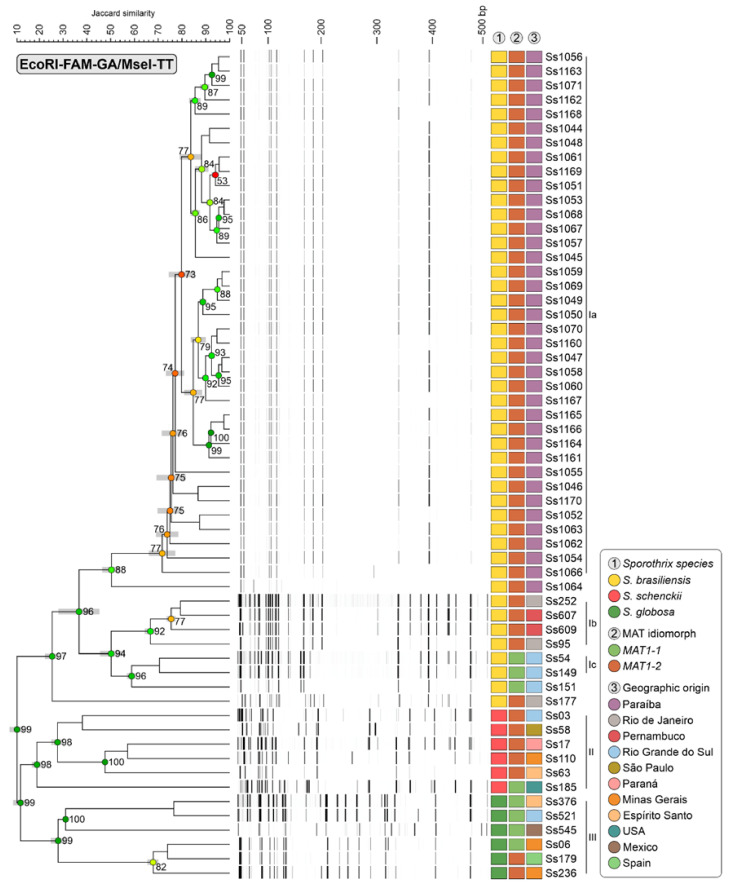
The dendrogram depicts the cluster profile generated by the AFLP method. The dataset consisted of 46 *S. brasiliensis* isolates (38 clinical isolates from Paraíba and 8 reference strains), 6 *S. schenckii* (reference strains), 6 *S. globosa* (reference strains). The dendrograms were constructed using the Jaccard similarity coefficient and UPGMA clustering algorithm in BioNumerics software v7.6.

**Figure 5 jof-10-00848-f005:**
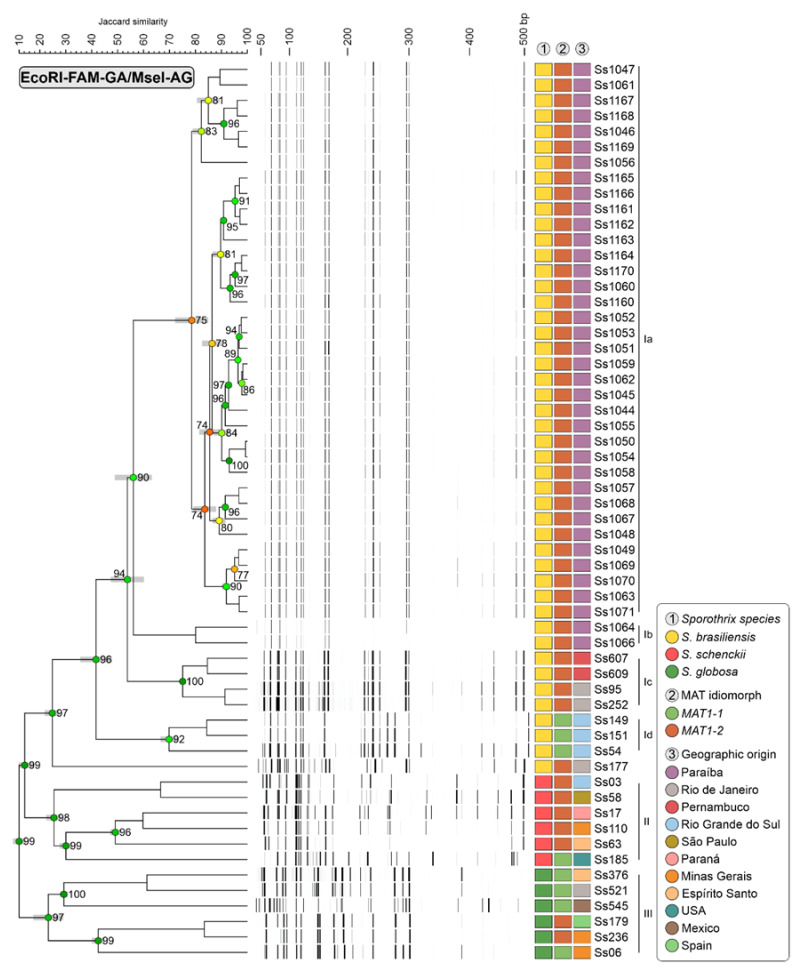
The dendrogram depicts the cluster profile generated by the AFLP method. The dataset consisted of 46 *S. brasiliensis* isolates (38 clinical isolates from Paraíba and 8 reference strains), 6 of *S. schenckii* (reference strains), 6 of *S. globosa* (reference strains). The dendrograms were constructed using the Jaccard similarity coefficient and UPGMA clustering algorithm in BioNumerics software v7.6.

**Figure 6 jof-10-00848-f006:**
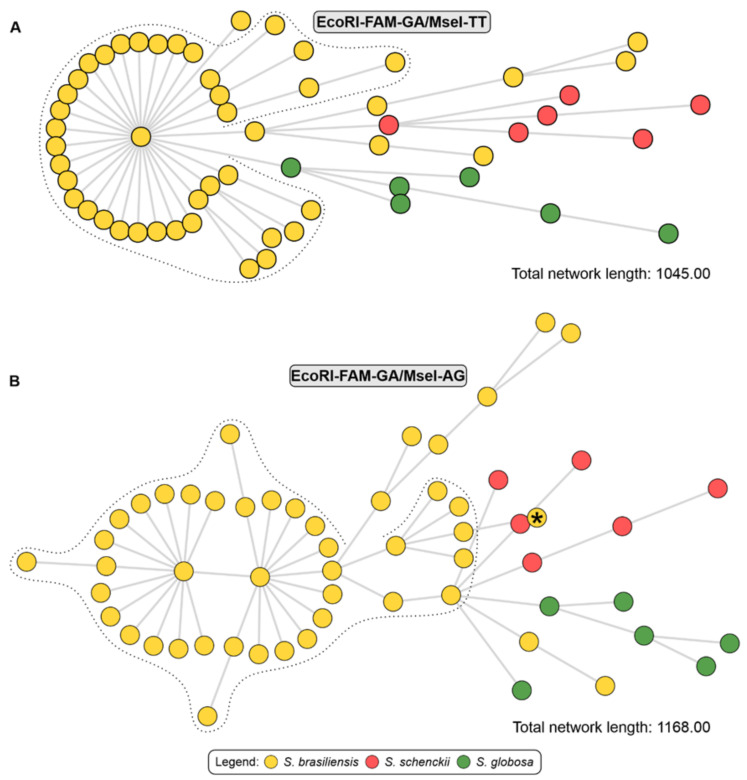
Minimum Spanning Tree (MST) derived from AFLP data. (**A**) In the #3 EcoRI-GA/MseI-TT combination and (**B**) In the #5 EcoRI-GA/MseI-AG combination. The dataset consisted of 46 *S. brasiliensis* isolates (38 clinical isolates from Paraíba and 8 reference strains), 6 of *S. schenckii* (reference strains), 6 of *S. globosa* (reference strains). The dotted lines represent the cluster profile obtained by the AFLP method of *S. brasiliensis* isolates from Paraíba. * Isolates from Paraíba. The analysis was performed using the BioNumerics software v7.6.

**Figure 7 jof-10-00848-f007:**
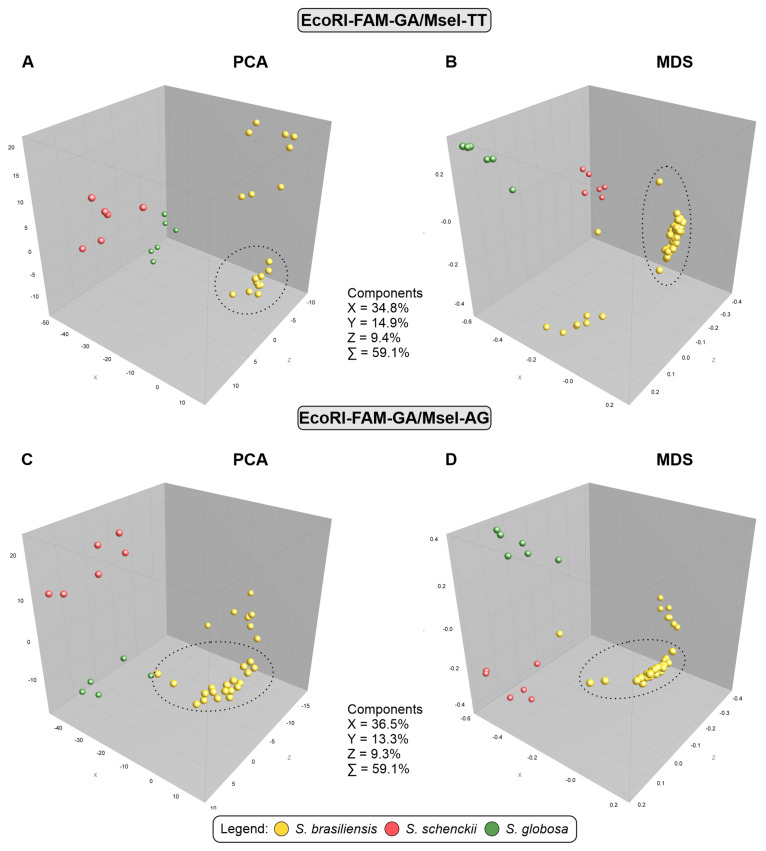
Principal component analysis (PCA) and multidimensional scaling (MDS). (**A**,**B**) In the #3 EcoRI-GA/MseI-TT combination and (**C**,**D**) In the #5 EcoRI-GA/MseI-AG combination The dataset consists of 46 isolates of *S. brasiliensis* (38 clinical isolates from Paraíba and eight reference strains), six of *S. schenckii* (reference strains), six of *S. globosa* (reference strains). The isolates were plotted in three-dimensional space and coloured according to genetic groups. The dotted lines represent the cluster profile obtained by the AFLP method of *S. brasiliensis* isolates from Paraíba. The analysis was performed using the BioNumerics software v7.6.

**Figure 8 jof-10-00848-f008:**
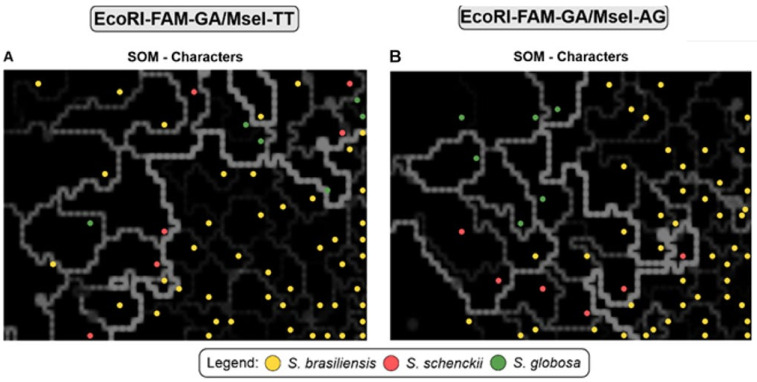
Distribution of *Sporothrix* spp. Genotypes generated by AFLP using a Self-Organizing Map (SOM). Kohonen maps using characters. (**A**) In the #3 EcoRI-GA/MseI-TT combination and (**B**) In the #5 EcoRI-GA/MseI-AG combination A total of 46 *S. brasiliensis* isolates (38 clinical isolates from Paraíba and eight reference strains), six of *S. schenckii* (reference strains) and six of *S. globosa* (reference strains) were included in this study. The dimensioning analyses were performed using BioNumerics v7.6 to assess the consistency of the differentiation of the populations defined by the cluster analysis. The distance between black blocks can be inferred by observing the thickness and brightness of the lines (white, grey) connecting them. The thicker and lighter the line, the greater the distance between the samples in the black blocks and their neighbouring blocks. Isolates were assigned specific colours corresponding to their genetic groups.

**Table 1 jof-10-00848-t001:** Clinical and epidemiological characteristics of patients with sporotrichosis included in this study (*n* = 36).

Variables	*N*	%
**Gender**		
Female	27	75%
Male	9	25%
**Clinical presentation**		
Lymphocutaneous	25	69%
Cutaneous-Fixed	11	31%
Disseminated	0	0%
**Source of infection**		
Zoonosis	34	94.4%
Environmental	2	5.6%

## Data Availability

The original contributions presented in the study are included in the article, further inquiries can be directed to the corresponding author.

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
