# Peer review of "Analyses of the Genetic Diversity and Population Structures of Sporothrix spp. Clinical Isolates from Paraíba, Brazil"

_jof, 2024, doi:10.3390/jof10120848_

Round 1
Reviewer 1 Report
In this work, the authors analyze the genetic diversity of clinical isolates of Sporothrix spp. from Paraíba, Brazil. Using molecular techniques, the authors determined that all clinical isolates belong to S. brasiliensis, with only a single sexual idiomorph detected. The AFLP results further suggest that one or two genetic groups may be circulating in specific areas of the region studied. Although the research focuses on a limited geographic area, it is significant as it enhances our understanding of how sporotrichosis spreads in a country where the infection is highly endemic, highlighting some specific differences in genetic divergence of S. brasiliensis compared to that found in other Sporothrix species.
To my knowledge, the study does not contain any methodological inaccuracies, and the experimental design is sound. The manuscript is well-structured, easy to follow, and provides sufficient background for the reader to grasp the purpose of the research. The methods are thoroughly described, and the results are generally presented clearly and concisely. The discussion offers sufficient depth and cites relevant references, enabling readers to contextualize the findings within the existing body of knowledge on the subject.
Specific comments:
Line 93: The acronym SM - ULAC has not been previously introduced, please define it.
Figure 1: For clarity, please indicate what the numbers next to the color code represent.
Table 1: The percentages indicated for the Source of infection do not correspond to the N values, please correct accordingly.
Figure 3: Panel B-II is missing a scale bar, and panel F-III lacks a label. Additionally, in the figure caption, please specify whether any staining was applied to the photomicrographs.
Table 2 (Supplementary): The table legend does not provide explanations for the column headings. Please revise to include the definitions or descriptions of each heading.
Figure 7: Please clarify in the figure legend what the dotted lines represent.
Make sure all scientific names are written in italics and standardize the spelling of Paraiba (sometimes it appears as Paraíba and sometimes as Paraiba)
Author Response
1 - Line 93: The acronym SM - ULAC has not been previously introduced, please define it.
R- SM – ULAC (mycology sector - Clinical Analysis Laboratory Unit). Added to manuscript.
2- Figure 1: For clarity, please indicate what the numbers next to the color code represent.
Table 1: The percentages indicated for the Source of infection do not correspond to the N values, please correct accordingly.
R- Added to manuscript.-
3- Figure 3: Panel B-II is missing a scale bar, and panel F-III lacks a label. Additionally, in the figure caption, please specify whether any staining was applied to the photomicrographs.
R- Added to manuscript.
4- Table 2 (Supplementary): The table legend does not provide explanations for the column headings. Please revise to include the definitions or descriptions of each heading.Table 2 (Supplementary): The table legend does not provide explanations for the column headings. Please revise to include the definitions or descriptions of each heading.
R- Supplemental description of the table. Attached is how i corrected it.

Reviewer 2 Report
My major concerns are that:
(1) The molecular identification method of strains.
(2) The numbers of the research samples.
Please see the detail comments for more information.
1. The authors use the superscript for the number of the cited literatures. To my knowledge about the journal, the numbers should be placed in a [ ] and not formatted as superscript.
2. In the first paragraph, the authors write: “The clade includes the following species: Sporothrix schenckii (sensu stricto), S. brasiliensis, S. globosa, S. luriei, and S. mexicana¹-².” Does it mean only the five species in the genus Sporothrix cause Sporotrichosis? Change Mexicana to mexicana.
3. In the introduction section, it could be better if a short paragraph about the genus Sporothrix is added demonstrating how many species is included and what species usually cause serious diseases.
4. In the “2.4.2. Species-specific PCR” section, the authors only use the strains of S. brasiliensis for identification, and then get the results that all the analyses strains belong to S. brasiliensis. For the molecular identification, why not sequence the pcr products and then do the blast on NCBI or perform phylogenetic analyses based on the sequences. This could be more accurate and is the commonly used method.
5. Italicize Sporothrix brasiliensis in line 208
6. The format of the literatures should be carefully checked and revised according to the instructions of the journal, as I see some aspects are not uniform.
7. My another main concern is that this study is based on a relatively small sample data from a region. This largely reduce the reliability of the research results. It could be better if the authors can accumulate samples and data from a large area and time interval. At least, the authors can fully discuss this aspect and make a propect of the future studies.
Author Response
2. In the first paragraph, the authors write: “The clade includes the following species: Sporothrix schenckii (sensu stricto), S. brasiliensis, S. globosa, S. luriei, and S. mexicana¹-².” Does it mean only the five species in the genus Sporothrix cause Sporotrichosis? Change Mexicana to mexicana.
R- It means that they are species of clinical relevance, which were grouped in this clade (pathogenic clade), demonstrating that they have a greater potential to cause human sporotrichosis. It really wasn't clear in the text, but I made a change.
4. In the “2.4.2. Species-specific PCR” section, the authors only use the strains of S. brasiliensis for identification, and then get the results that all the analyses strains belong to S. brasiliensis. For the molecular identification, why not sequence the pcr products and then do the blast on NCBI or perform phylogenetic analyses based on the sequences. This could be more accurate and is the commonly used method.
R- Not only strains of S. brasiliensis were used for identification, what we used was a reference strain of S. brasiliensis as a positive control of the evidence, but species-specific PCR was performed on all proven samples. If any were not identified as S. brasiliensis, a specific PCR would be carried out for the other species.
7. My another main concern is that this study is based on a relatively small sample data from a region. This largely reduce the reliability of the research results. It could be better if the authors can accumulate samples and data from a large area and time interval. At least, the authors can fully discuss this aspect and make a propect of the future studies.
R- On the line 394-397 discuss this aspect and make a propect of the future studies.
''Considering that the study samples were of convenience, covering only the capital and metropolitan regions and the years 2019 to 2020, future studies that expanded the coverage area and period of the study would be important to consolidate this genetic characterization of Sporothrix in Paraíba.''
Although the research focuses on a limited geographic area, it is significant as it enhances our understanding of how sporotrichosis spreads in a country where the infection is highly endemic, highlighting some specific differences in genetic divergence of S. brasiliensis.
Round 2
Reviewer 2 Report
The manuscript was slightly revised. My concerns and comments are responsed.
The manuscript was improved though some questions can not be resolved at this stage.